# Multidistrict Host–Pathogen Interaction during COVID-19 and the Development Post-Infection Chronic Inflammation

**DOI:** 10.3390/pathogens11101198

**Published:** 2022-10-18

**Authors:** Marialaura Fanelli, Vita Petrone, Margherita Buonifacio, Elisabetta Delibato, Emanuela Balestrieri, Sandro Grelli, Antonella Minutolo, Claudia Matteucci

**Affiliations:** 1Department of Experimental Medicine, University of Rome Tor Vergata, 00133 Rome, Italy; 2Department of Food Safety, Nutrition and Veterinary Public Health, Istituto Superiore di Sanità, 00161 Rome, Italy; 3Virology Unit, Tor Vergata University Hospital, 00133 Rome, Italy

**Keywords:** COVID-19, post-COVID-19 syndrome, SARS-CoV-2, microbial triggers, chronic inflammation, microbiota, immunological dysfunction

## Abstract

Due to the presence of the ACE2 receptor in different tissues (nasopharynx, lung, nervous tissue, intestine, liver), the COVID-19 disease involves several organs in our bodies. SARS-CoV-2 is able to infect different cell types, spreading to different districts. In the host, an uncontrolled and altered immunological response is triggered, leading to cytokine storm, lymphopenia, and cellular exhaustion. Hence, respiratory distress syndrome (ARDS) and systemic multi-organ dysfunction syndrome (MODS) are established. This scenario is also reflected in the composition of the microbiota, the balance of which is regulated by the interaction with the immune system. A change in microbial diversity has been demonstrated in COVID-19 patients compared with healthy donors, with an increase in potentially pathogenic microbial genera. In addition to other symptoms, particularly neurological, the occurrence of dysbiosis persists after the SARS-CoV-2 infection, characterizing the post-acute COVID syndrome. This review will describe and contextualize the role of the immune system in unbalance and dysbiosis during SARS-CoV-2 infection, from the acute phase to the post-COVID-19 phase. Considering the tight relationship between the immune system and the gut–brain axis, the analysis of new, multidistrict parameters should be aimed at understanding and addressing chronic multisystem dysfunction related to COVID-19.

## 1. COVID-19: From Infection to Host Antiviral Response

### 1.1. SARS-CoV-2 Infection and COVID-19 Disease

In December 2019, the Wuhan Municipal Health Commission (China) reported to the World Health Organization (WHO) a cluster of pneumonia cases of unknown etiology in Wuhan, Hubei province [1]. The coronavirus identified in Wuhan is a new viral strain that has not previously been identified in humans, namely, SARS-CoV-2. The SARS-CoV-2 genome is composed of six functional open reading frames (ORFs) arranged in order from 5′ to 3′: replicase (ORF1a/ORF1b), spike (S), envelope (E), membrane (M), and nucleocapsid (N) [2]. The S glycoprotein spike is essential for the internalization of SARS in the host cell. It consists of two main components: the S1 and S2 subunits. The S1 subunit contains the receptor-binding domain (RBD), through which the virus is recognized and binds the angiotensin-converting enzyme 2 receptor (ACE2), which is expressed in host cells [3]. The S2 subunit consists of a specific conserved amino acid sequence called fusion peptide (FP), which is involved in the fusion of the virus to the host cell membrane. For the adhesion to ACE2, the S protein must undergo cleavage by specific proteases that vary according to cell type [4]. Among the key transmembrane proteases is the serine protease transmembrane protease serine 2 (TMPRSS2), which is structurally and functionally associated with the ACE2 receptor. TMPRSS2 acts on the S protein at the S1/S2 cutoff site, detaching the S1 subunit and thereby ensuring fusion with the cell membrane via the S2 subunit [5]. After the virus enters the host cell, the viral genome is released into the cytoplasm, and translation and replication occur, resulting in the formation of precursor genomes and sub-genomic mRNAs that are transformed into membrane proteins, N-proteins, and various accessory proteins [6]. All these mechanisms play a key role in human transmissibility and the pathogenesis of the coronavirus.

The World Health Organization identified COronaVIrus 19 disease (COVID-19), the illness caused by SARS-CoV-2. Fever, shortness of breath, dry cough, myalgia, and fatigue are the main clinical manifestations, but other symptoms such as headache, rhinorrhea, sneezing, sore throat, loss of smell, and pneumonia are associated with COVID-19 [7]. Although most cases lead to complete recovery, SARS-CoV-2 can cause severe infection and result in septic shock and acute respiratory distress syndrome (ARDS), as well as acute cardiac damage, acute renal damage, and multi-organ failure, requiring admission to intensive care units [8]. Extremely severe cases, which constitute about 10 percent of the infected population, can lead to death [8]. In children, the illness is usually mild. Cardiovascular disease, lung disease, kidney disease, or neoplasia are the primary comorbid causes of severe COVID-19 [9].

Moreover, genetic factors are quite important too. Some polymorphisms that are common in the general population and that affect the functionality, or the expression of the protein encoded by the mutated gene are associated with severe COVID-19. In many cases, this observation confirms the role of the altered host protein in the pathogenesis of COVID-19 [10,11,12]. For example, regarding the mechanism of virus entry, cellular protease TMPRSS2 polymorphisms were found to be associated with the risk of COVID-19 [13]. In addition, reduced type I interferon functionality has been demonstrated in patients with a severe form of COVID-19. This may be due either to a genetic mutation or to the presence of auto-antibodies. Using a genetic approach, mutations were identified in genes involved in the regulation of type I and type III IFN immunity that may underlie an inadequate response and be implicated in severe COVID-19 pneumonia [14]. Furthermore, high titers of neutralizing autoantibodies against IFN-α2 and type I IFN-ω have been detected in about 10% of patients with severe COVID-19 pneumonia, suggesting their ability, in this case, to cause direct host damage rather than protection against infection [15,16].

The lung is the main organ involved in COVID-19, the impairment of which is the main cause of death. Hypoxic respiratory failure occurs, resulting from acute respiratory distress syndrome, which often needs assisted ventilation. The main features of lung impairment are alveolar epithelial cell damage, cytokine storm, microvascular endothelial damage, and thrombosis [17]. However, less common symptoms such as diarrhea, nausea, vomiting, and abdominal pain are found with very variable frequency in different populations, presenting in early and mild forms, often followed by the typical respiratory syndrome [18].

Liver damage with increased transaminases, hypoproteinemia, and prolonged prothrombin time have been found in COVID-19 patients. Usually, hepatotoxicity is caused by viral hepatitis, a side effect due to drug treatments, or an overreaction of the immune system. Limited information is available on whether SARS-CoV-2 infects liver cells directly. Recent data have shown a significant increase in ACE2 expression in cholangiocytes instead of hepatocytes, suggesting that COVID-19 could lead to direct damage to the intrahepatic bile ducts [19,20].

Evidence of intestinal involvement is not only the gastrointestinal symptoms but also the presence of viral RNA in the patient’s stool. Numerous studies have demonstrated the presence of viral RNA in the stool or anal/rectal swabs of patients with COVID-19. Therefore, the gastrointestinal virus persistence and the presence in feces highlights that SARS-CoV-2 is not limited to the lungs and indicates a potential fecal–oral transmission that can occur even after viral clearance in the respiratory tract [21].

In addition to respiratory, vascular, and gastrointestinal symptoms, neurological symptoms have also been observed in the acute and post-acute phases. The National Institutes of Health (NIH) reports neurological symptoms in acute COVID-19, including loss of taste and smell, headache, stroke, delirium, and brain inflammation. It has also been reported that the effects of COVID-19 can persist long after infection. Brain fog, fatigue, headache, dizziness, and shortness of breath have mainly been identified and are called post-acute sequelae of COVID-19 (or PASC). Notably, patients who have been hospitalized are the most susceptible; even those with mild cases may also experience long COVID due to the persistence of SARS-CoV-2 infection [22,23].

### 1.2. Multi-Systemic Alteration Due to SARS-CoV-2 Infection

The ability of SARS-CoV-2 to induce significant multi-organ dysfunction in COVID-19 patients is well known, in addition to the more pronounced symptoms due to lung and airway impairment. To understand and explain the virus’s tropism toward different tissues, it is important to focus on the interaction between SARS-CoV-2 and the ACE2 receptor, through which the virus enters and infects host cells [3]. ACE2 is present in different tissues, with the highest expression in the small intestine, testes, kidney, heart, thyroid, and adipose tissue; intermediate levels have been found in the lungs, colon, liver, bladder, and adrenal gland, and low levels have been found in blood, spleen, bone marrow, brain, blood vessels, and muscles [20]. In addition, a protein profile of ACE2 has been characterized by immunohistochemistry assays, which allowed for the localization to be defined, confirming high expression in the intestinal tract, more abundance in the duodenum and small intestine, and lower levels in the stomach [24]. These results show that ACE2 is expressed in different human tissues and suggest how its heterogeneous expression could explain the multisystem impairment associated not only with the cytokine storm triggered by our immune systems but also with the direct infection of SARS-CoV-2 in different organs. Indeed, Liu and colleagues confirmed the co-localization of SARS-CoV-2 spike proteins and ACE2 and also demonstrated the presence in different tissues such as the lung, trachea, small intestine, and kidney [25].

Moreover, a recent study showed the presence of SARS-CoV-2 RNA in the cardiac tissue of 41 cases (43%) out of 95 SARS-CoV-2-positive deceased and how the infection in the heart induces transcriptomic alterations associated with the immune response and an increase in pro-inflammatory gene expression [26]. Wanner and colleagues demonstrated, using autopsy specimens, SARS-CoV-2 liver tropism in 69% of COVID-19 patients, confirming abnormal liver function via viral RNA detection and viruses isolated from post-mortem liver tissue [27].

Schaller and co-authors developed a system for the ex vivo study of human SARS-CoV-2 infections using the biopsies of cryopreserved human lung tissues, and through the in situ hybridization of viral RNA and quantitative PCR, it was possible to demonstrate the SARS-CoV-2 infection of lung cells [28].

The intestinal tract, given the numerous symptoms found in COVID-19 patients, is also considered a possible target organ for SARS-CoV-2. Lehmann and co-authors demonstrate the presence of SARS-CoV-2 RNA and nucleocapsid protein in the duodenal mucosa, associated with a morphological change in the epithelium combined with an accumulation of activated CD8 T-cells [29].

The presence of SARS-CoV-2 in nerve tissue has also recently been documented and demonstrated by the neurological symptoms observed in the vast majority of individuals with COVID-19. Melo and colleagues analyzed the olfactory mucosa in COVID-19 patients who reported a loss of olfaction, demonstrating the presence of SARS-CoV-2 viral particles and inflammation in multiple cell types in the olfactory neuroepithelium, including olfactory sensory neurons [30]. Meinhardt and colleagues demonstrated the presence of SARS-CoV-2 RNA and protein in anatomically distinct regions of the nasopharynx and brain, presenting evidence of SARS-CoV-2 neurotropism [31]. SARS-CoV-2 has also been shown to be present in defined neuroanatomical areas including the primary respiratory and cardiovascular control center in the medulla.

### 1.3. Immunopathology of COVID-19 Disease

An elevated inflammatory response and immune dysregulation are the main consequences of SARS-CoV-2 infection and underlie COVID-19 disease. This infection results in the activation of macrophages and dendritic cells (DCs) that trigger an initial immune response, including lymphocytosis and cytokine release. The SARS-CoV-2 virus, via specific, extracellular binding to the ACE2 receptor, is internalized and confined to the endosome, inducing a significant innate immune response through the host’s recognition of several viral components [4,5,8].

#### 1.3.1. The Activation of the Innate Immune Response by Viral-Associated Molecular Patterns

During an innate immune response to a viral infection, different molecular structures that are characteristic of virus-defined pathogen-associated molecular patterns (PAMPs) are recognized by pattern recognition receptors (PRRs). Based on cellular localization, PRRs are distinguished as Toll-like (TLR) or RIG-I-like (RLR). TLRs are located on the cell surface (TLRs 1, 2, 4, 5, 6) or bound to intracellular endosomal membranes (TLRs 3, 7, 8, 9), whereas RLRs are located at the cytoplasmic or nuclear level [32]. The RLR family includes retinoid acid-inducible gene I (RIG-I) and melanoma differentiation factor 5 (MDA5), which, through RNA helicase activity, can trigger the antibody response by activating the mitochondrial antiviral signaling protein (MAVS) [33].

The recognition of single-stranded RNA (ssRNA) by TLR7/8 occurs at the endosomal level, leading to the activation of the MyD88 pathway, which mediates the activation of NF-κB and interferon regulatory factor-3 (IRF-3) and 7 (IRF-7). This results in the transcription of proinflammatory cytokines and type I and III interferons (IFNs) [34,35]. During viral replication of SARS-CoV-2, double-stranded (ds) RNA is also produced, which binds TLR3 and activates TIR domain-containing adapter protein, inducing the interferon β (TRIF) pathway. Considering the protein components of SARS-CoV-2, the ability of TLR2 to bind the E (envelope) protein and the subsequent activation of MyD88 in association with inflammation in the lung district have also been demonstrated [36,37]. The combination of different viral proteins—for example, those released from endosomes and those translated by reading viral genomic RNA—are recognized by RLRs that associate with MDA-5 and MAVS and activate IRF3/7, leading to the secretion of IFN-α/β [38,39]. The binding of PAMPs to PRRs triggers the initiation of an inflammatory response against the virus, resulting in the activation of several signaling pathways and, subsequently, transcription factors. The ‘cytokine storm’ is the result of a sudden increase in the circulating levels of several pro-inflammatory cytokines, including IL-6, IL-1β, TNF-α, and interferon. This triggers an altered inflammatory response that can lead to respiratory distress syndrome (ARDS) and systemic multi-organ dysfunction syndrome (MODS) [40] (Figure 1).

#### 1.3.2. The SARS-CoV-2-Induced Activation of the IFN System

The first phase of infection caused by microbial infection is characterized by the activation of mononuclear phagocytes (MPs). Monocytes, macrophages, and dendritic cells are protagonists that regulate the first host–pathogen interaction and are referred to as antigen-presenting cells (APC). They are heterogeneous populations, and, in addition to their role in innate immunity against pathogens, they have a role in inflammation and its resolution, wound healing and tissue repair, maintaining homeostasis, and regulating interactions between the microbiota [41]. The first line of defense is the production of interferons by APC cells and also by cells involved in viral infection. One study showed that the in vitro stimulation of human PBMCs by SARS-CoV-2 induces TLR7/8-dependent cytokine and chemokine production, in particular, a type I and type III IFN response. The same group showed that the cells responsible for the production of type I interferons, such as IFN-αs and IFN-β, during infection are plasmacytoid dendritic cells (pDCs) [42].

During the early phase of viral infection, interferons can regulate inflammation and the immune response. There are three classes of IFNs: type I (FN-α and IFN-β, IFN-ε, IFN-κ, and IFN-ω [42,43,44,45,46]); type II (IFN-γ, produced by activated T cells, natural killer cells, and macrophages [47]); and type III (IFN-λ1, IFN-λ2, IFN-λ3, and IFN-λ4). IFN III (IFN-λ) responses are mainly restricted to mucosal surfaces and confer antiviral protection [48]. Type I IFN and type III IFN are involved in the antiviral response and represent the connection between innate and adaptive immunity. The antiviral IFN system comprises an initial induction phase, characterized by production through the activation of transcription factors, and a phase called IFN signaling [49,50]. The released IFNs act in both an autocrine and paracrine manner by binding IFN receptors on the cell surface. This mechanism of action leads to the activation of IFN-stimulated gene factor 3 (ISGF3), which translocates to the nucleus and binds to interferon-stimulated response elements (ISRE), leading to further amplification and development of immune responses [14,51].

The IFN-mediated response is crucial during the early phase of SARS-CoV-2 infection and has also been shown to be closely involved with the evolution of the disease. Different research groups have shown that genetics is crucial. Zhang and colleagues identified patients with severe COVID-19 who had mutations in genes involved in the regulation of type I and III IFN immunity [14]. Bastard and colleagues identified individuals with high neutralizing autoantibody titers against IFN-α2 type I and IFN-ω in approximately 10% of patients with severe COVID-19 pneumonia. These autoantibodies were not found in asymptomatic and mild patients or in healthy individuals [52,53]. In a recent study, Simula and co-authors found higher levels of autoantibodies against IFN-α and IFN-ω in ICU patients with life-threatening COVID-19 than in subjects with mild COVID-19 and healthy subjects [54]. In another work, it was shown that congenital errors in TLR3 and IRF7-dependent type I IFN immunity underlie critical COVID-19, demonstrating the essential role of both the TLR-3 double-stranded RNA sensor and intrinsic immunity to type I IFNs in controlling SARS-CoV-2 infections in the lungs. In addition, autosomal-recessive deficiencies in IRF7 and IFNAR1 were also found [14,53]. Another important function of type I IFNs is to induce the expression of an adhesion molecule known as CD169/SIGLEC1. CD169 is a member of the sialic acid-binding immunoglobulin-like lectin (SIGLEC) family and plays an important role in the course of viral infections, including those caused by the Ebola virus and human immunodeficiency virus (HIV) [55]. It has been shown that CD169 is expressed on the surface of DCs and monocytes after the release of antiviral molecules, and its expression on monocytes has been correlated with IFN type I levels [56,57]. Its role as an early marker of viral infection has been investigated in the course of SARS-CoV-2 infection. Bourgoin and colleagues showed that CD169 expression on monocytes was elevated in 95% of patients with SARS-CoV-2 infection compared with normal levels [58]. Doehn and co-authors demonstrated that CD169 expression is associated with disease severity [59]. Our group demonstrated the early expression of CD169 in COVID-19 patients correlated with senescence and exhaustion markers in the CD8 T cell subset, as well as with B cell maturation and differentiation markers [60]. CD169 RMFI also reflected the severity and the respiratory outcome of COVID-19 patients during hospitalization. In addition, stimulation with the SARS-CoV-2 spike protein was able to trigger CD169 in PBMCs in a dose-dependent way, in association with the increased transcription of IL-6 and IL-10 genes [60] (Figure 1).

#### 1.3.3. Adaptive Immune Dysregulation and Hyper Inflammation in COVID-19

The pathological picture of patients with COVID-19 is characterized by lymphopenia, a relative increase in neutrophils, the depletion of CD8-positive T cells, a rise in Th17, and a decrease in Treg cell responses, which have been associated with hyperinflammation and subsequent cytokine storms [61]. The inefficient and uncontrolled initial response is then reflected at the systemic level. At the cellular level, an important lymphopenia was observed specifically in T cells associated with disease severity. Based on the literature, the causes of lymphopenia could be excessive inflammation, in particular, due to high levels of the cytokine IL-6 and the downregulation of genes involved in T cell expansion, such as MAP2K7 and SOS1 caused by SARS-CoV-2 infection [62]. In a recent study, the presence of leukocytes in the ceca and colons of COVID-19 patients was demonstrated using white blood cell (WBC) scans, clarifying how lymphocytes recalled to peripheral lymphoid tissues in the gut could be involved in lymphopenia [63].

Normally, T cells act as regulators of humoral and adaptive immunity: Both CD4+ and CD8+ lymphocytes recognize MHC-associated pathogen peptides through their receptors (TCR), and this mechanism leads to differentiation. In viral infections, CD8+ lymphocytes block infected cells from producing effector molecules, such as granzyme A/B and perforin, or through CD95/Fas-mediated apoptosis; CD4+ follicular helper lymphocytes (Tfh) stimulate B lymphocytes to produce specific antibodies [64], but lymphopenia does not allow for the proper function of antiviral T cell activity.

Indeed, a significant change in the differentiation phenotype—with an increase in terminal effector memory T cells and a decrease in naïve and early memory T cells, as well as with the elevated expression of the senescence marker CD5—could explain this dysfunction in the immunological response [60,65]. Moreover, an increase in PD-1 (programmed cell death-1), which regulates cell exhaustion, has been found in CD4+ and CD8+ cells [66,67,68,69,70].

This persistent inflammatory state and inadequate immune response, characteristic of the acute phase of COVID-19 disease, may persist over time and cause post-acute COVID syndrome. Patients with long COVID showed significant immune activation, a reduced proportion of naïve T and B cells, and a high expression of IFN-β and IFN-λ1 [71].

COVID-19 patients show high levels of pro-inflammatory cytokines and chemokines, such as IFN-γ, TNF-α, IL-6, IL-1β, IL-18, CXCL8, and CXCL10, and the anti-inflammatory and regulatory cytokines (IL-4, IL-7, IL-10, and TGFβ) responsible for early immune response mechanisms and cytokine storm in the post-infection phase [72]. Uncontrolled systemic inflammatory processes occur due to the actions of the cells of innate immunity, such as neutrophils, macrophages, and NK cells. Neutrophils produce neutrophil extracellular traps (NETs), networks of extracellular reticular fibrillar structures that contain the infection, which, when dysregulated, can induce thrombus formation, and amplify cytokine release [73]. Macrophages, stimulated by the inflammatory environment, become activated and release large amounts of cytokines, contributing to tissue damage [74]. The elevated presence of IL-6 and IL-10 also affects the cytolytic function of NK cells, key mediators of antiviral response, decreasing the expression of perforin and granzyme B in NK cells [75,76] (Figure 1).

## 2. The Role of Microbiota in Health and Disease

The important relationship between the host and microorganisms, known as the holobiont, has consolidated over time, and has become a research focus [77]. The microbiota is composed of bacteria, viruses, and fungi and contributes to both healthy and pathological conditions [78]. The evolution between humans and microorganisms has led to the development of microbial niches specific to each body district [79]. Our bodies have learned to maintain a close and balanced relationship with the microbiota, which has evolved over time as a form of symbiosis [80]. For instance, in the gut, this interaction allows the body to obtain benefits, such as the production of important nutrients, including vitamins and short-chain fatty acids (SCFA) [81]. Moreover, mucosal-associated lymphoid tissue (MALT) plays an anti-infective and barrier role, an immunoregulatory function, and regulates immunological tolerance. The mucous membranes represent not only the site through which infection occurs but also an element in close contact with the microorganisms residing in our bodies. The disruption of the balance between the host and the microbiota, which may be due to a new infection or the impairment of the immune system, induces a dysregulated immune response in the mucosa, thus leading to inflammatory processes underlying various diseases [82,83].

### 2.1. The Different Districts of the Mucosal Immune System

Mucosal surfaces are characterized by the presence of lymphoid cells that are part of a MALT, which, according to the different areas, becomes gut-associated lymphoid tissues (GALTs) in the intestine [84], bronchus-associated lymphoid tissue (BALT) in the lungs [85], skin-associated lymphoid tissue (SALT) in the skin [86], nasal-associated lymphoid tissues (NALT) in the nose [87], larynx-associated lymphoid tissue (LALT) in the larynx, and conjunctiva-associated lymphoid tissue (CALT) in the conjunctiva [88]. The role of the mucosal immune system is to protect the body’s internal surfaces while keeping the physiological functions of the relevant organs intact and efficient, such as food absorption, gas exchange, and sensory and reproductive activities [89].

GALTs have a compartmental structure typical of peripheral lymphoid organs and are the sites where peripheral immune responses occur. The cells of the immune system (macrophages, lymphocytes, dendritic cells) are found throughout the digestive tract in organized tissues, at the level of the mucosa and lamina propria. GALTs include Peyer plaques in the small intestine and isolated lymphoid follicles (ILFs) in the walls of the entire intestine. Microbe-associated molecular profiles (MAMPs), detected by PRRs on intestinal epithelial cells (IECs) and DCs, stimulate B cell recruitment and the subsequent maturation of ILFs, which can produce IgA through T-dependent and -independent interactions. Due to the increased transportation of microbes and their products through the epithelium, they enter Peyer plaques and ILFs via M cells before being endocytosed by dendritic cells in the subepithelial dome. This induces T cell differentiation, the maturation of T-dependent B cells, the subsequent release of dimeric IgA into the intestinal lumen, and the increased activation of lamina propria DCs that migrate through afferent lymphatic vessels to a draining MLN. IECs proliferate in the crypts, and Paneth cells and their components increase in density for the release of AMPs (antimicrobial peptides) [84,90].

### 2.2. Microbiota Alteration during Respiratory Virus Infections

The respiratory tract turns out to be the first site that most viruses use for the transmission of the most common respiratory viruses (enterovirus, influenza virus, adenovirus, respiratory syncytial virus) and coronaviruses [91,92]. The microbiota of the upper respiratory tract (oropharynx) is characterized by the presence of *Actinobacteria*, *Firmicutes*, *Proteobacteria*, and *Bacteroidetes*, while at the lung level, *Prevotella*, *Veillonella*, and *Streptococcus* are present. Furthermore, there are mainly *Bacteroides*, *Faecalibacterium*, and *Bifidobacterium* at the intestinal level [93]. The respiratory and gut microbiota play a key role during major airborne infections in humans. Few studies have demonstrated microbiota changes in prevalent respiratory viral infections. A study showed the presence of *Fusicatenibacter*, *Romboutsia*, *Anaerostipes*, *E. hallii*, *Ruminococcus torques*, and *Blautia* in H1N1-infected patients compared with healthy donors [94], as well as the excessive growth of *Escherichia coli* and *Enterococcus faecium*, and an important reduction in diversity was found during H7N9 infection [95]. Several studies have shown a strict association between the lung and gut, defined as the lung–gut axis, by contextualizing their interplay in various alterations in which the microbiota is involved [96].

### 2.3. The Contribution of Microbiota to COVID-19 Disease

The microbiota is a master regulator of the host’s homeostasis, suggesting it could play a key role in the regulation of hyperinflammation induced by SARS-CoV-2 infection and could contribute to COVID-19 disease severity. The microbiota plays a fundamental role in the modulation, formation, and function of the host immune system; given the multitude of microbes present in the gut, the immune system has evolved not only to induce protective responses against pathogens but also to establish regulatory pathways involved in maintaining tolerance to harmless antigens [81]. A highly regulated symbiotic relationship has been established between the host and the gut microbiota, which is sensitive to changes caused by diet, drug use, and pathogenic infections. The transport, processing, and presentation of foreign antigens, as well as the induction and clonal expansion of effector lymphocytes, occur at specific sites in the mucosa that are characterized by the presence of epithelium-specific organized lymphoid tissue [80,82].

#### Microbiota Alterations in Oral Cavity and Nasopharynx during COVID-19

The oral cavity includes the second largest community of the human body microbiota, and the upper respiratory tract is the main entrance for SARS-CoV-2 due to the transmission by droplets and aerosols [97,98,99]. The nasopharyngeal microbiota from mild COVID-19 patients compared to uninfected controls was found to be like controls, suggesting a resilient microbiota in early mild COVID-19.

The main phyla detected in samples were *Firmicutes*, *Bacteroidetes*, *Proteobacteria*, *Actinobacteria*, and *Fusobacteria* [100]. In a pre-print work, Budding and co-authors showed two different microbiota clustering in the pharyngeal microbiota of COVID-19 patients compared with controls. A heterogeneous cluster characterizes positive SARS-CoV-2 samples, while a homogenous microbiota cluster represents negative samples. Moreover, in older patients, microbial diversity decreases, suggesting an age-dependent pharyngeal dysbiosis and susceptibility to SARS-CoV-2 infection [100].

Nasopharyngeal commensal bacteria, including *Gemella morbillorum*, *Gemella haemolysans*, and *Leptotrichia hofstadii*, reduce in the pharynx of COVID-19 patients, while an increase in *Prevotella histicola*, *Streptococcus sanguinis*, and *Veillonella dispar* can be observed. Furthermore, the elevated presence of *G. haemolysans* and *L. hofstadii* is significantly positively associated with serum chlorogenic acid methyl ester, which could be an anti-SARS-CoV-2 bacterial metabolite [101].

### 2.4. Lung Microbiota during COVID-19

Healthy human lungs present with the following dominant genera: *Prevetella*, *Veillenella*, *Pseudemenes Fusobacteria*, and *Streptococcus*. [102]. Fan and colleagues analyzed lung microbiota in biopsies from fatal cases of COVID-19; in these patients, *Acinetobacter*, *Brevundimonas*, *Burkholderia*, *Chryseobacterium*, *Sphingobium* species, and *Enterobacteriaceae* members predominated the lung microbiota. In the lungs of deceased COVID-19 patients, the Enterobacteriaceae family was detected, which includes some pathogenic microbes, such as *Enterobacter*, *Escherichia coli*, *Klebsiella*, and *Proteus*, usually present in intestinal microbiota [103]. In addition, the ‘gut–lung axis’ relationship plays a role in contrasting the ‘cytokine storm’, a detrimental characteristic of COVID-19. Patients with COVID-19 pneumonia undergoing oral bacteriotherapy as a complementary therapy have shown a consistent reduction in mortality and more effective symptom control [104]. This finding highlights the possibility of crosstalk or migration from the gut to the lung microbiome. These mechanisms could be investigated, but one possible hypothesis is that endotoxin is secreted by pathogenic *Enterobacteriaceae* in gut and lung epithelial cells, leading to emphasized pulmonary inflammation. Moreover, *Acinetobacter baumannii* is associated with multiresistant infections and mortality [105]. The analysis of lung fungal microbiota in COVID-19 patients is dominated by *Cryptococcus*, which is related to high mortality in immunocompromised individuals. The opportunistic species *Cryptococcus*, *Issatchenkia*, *Cladosporium*, and *Candida* also are involved in mycosis in immunosuppressed patients [102,106].

### 2.5. Changes in Intestinal Microbiota in COVID-19 Patients

The gut microbiota has an important impact on health and disease considering its involvement in nutrient absorption and metabolism [107]. These functions are regulated by the interplay between the immune system and changes in the microbiota composition induced by exogenous infection, including those of viral origin, which could result in its disruption. For instance, gastrointestinal disorders are common during COVID-19. Gu and colleagues demonstrated the different gut microbial characteristics of patients with COVID-19, H1N1-infected patients, and healthy controls. In particular, the SARS-CoV-2-infected patients had a decrease in the diversity of the intestinal microbiota when compared with controls and with respect to H1N1 patients, with a predominance of opportunistic genera, such as *Actinomyces*, *Rothia*, *Streptococcus*, and *Veillonella*, as well as a decrease in the relative abundance of *Bifidobacterium* genera [94]. Zuo and colleagues studied changes in the fecal microbiota of 15 patients with SARS-CoV-2 infection during hospitalization and its association with the severity of the disease. The fecal microbiota of COVID-19 patients showed significant alterations, with a prevalence of opportunistic pathogens such as *Clostridium hathewayi*, *Actinomyces viscous*, and *Bacteroides nordii* and a depletion in symbiotic bacteria useful for host immunity (*Faecalibacterium prausnitzii*, *Lachnospiraceae bacterium*, *Eubacterium rectale*, *Ruminococcus obeum*, and *Doreaformicigenerans*).

On this basis, the microbiota could play a pivotal role as a biomarker of susceptibility to SARS-CoV-2 infection [108]. These changes may improve or persist even after the symptoms have resolved and the virus has been eliminated [109]. The severity of COVID-19 correlates positively with the relative abundance of *Coprobacillus*, *Clostridium ramosum*, and *Clostridium Hatheway*, while the abundance *of Faecalibacterium prausnitzii*, associated with an anti-inflammatory microenvironment, is inversely correlated to the severity of the disease [110,111]. Moreover, greater interindividual fecal mycobiome variation in COVID-19 patients compared with healthy controls has been demonstrated. In particular, it was seen that, during all periods of hospitalization, patients with SARS-CoV-2 infection show an increase in opportunistic fungi, including *Candida albicans*, *C. auris*, *Aspergillus flavus*, and *A. niger*. In fecal samples, the latter two respiratory pathogens were detected even after SARS-CoV-2 clearance and the resolution of respiratory symptoms, suggesting unstable intestinal mycobiomes and persistent dysbiosis in some COVID-19 patients [112].

Recently, research has shown that, as in adults, the intestinal microbiota of children with COVID-19 are rich in bacteria with predominantly pro-inflammatory properties (Bacteroidetes and Fusobacteria) and poor in certain microorganisms (*Akkermansia*, *Blautia*, *Ruminococcus*) that favor the maintenance of intestinal balance (homeostasis). A significant increase in *Faecalibacterium*, a bacterium known for its beneficial and anti-inflammatory properties, supports the immune system in defending the body and is also absent in adult patients with the more severe form of COVID-19. These analyses support the hypothesis that child microbiota, due to anti-inflammatory properties, could reduce the severity of infection [113], suggesting a contributing factor to the different age-related responses to SARS-CoV-2 infection.

## 3. Post-Acute COVID-19 Syndrome

SARS-CoV-2 infections cause several alterations characteristic of COVID-19. Considering the significant impact on various body districts important and persistent symptoms following infection are now known. Complex symptoms lasting four or more weeks after infection are defined by the Centers for Disease Control and Prevention (CDC) as ‘post-COVID conditions’ (PCC), which are characterized by physical, social, and psychological impairments [114,115,116]. Different terms are commonly used to define these persistent post-infection symptoms. Symptoms may arise from the acute phase of COVID-19, be newly onset, or evolutions of pre-existing ones. Post-COVID conditions are defined by different and heterogeneous symptoms that may appear together or alone: neurological, psychiatric, cutaneous, gastrointestinal, systemic, and cardiorespiratory [117,118,119] (Figure 2).

Considering the impact of these symptoms on the population after infection with SARS-CoV-2, the post-COVID condition was incorporated into the International Classification of Diseases, Tenth Edition Clinical Modification (ICD-10-CM), published by the World Health Organization [120].

The alterations that appear post-infection could be caused by a persistent chronic inflammatory state. It has recently demonstrated that cytokines such as IL-1b, IL-6, TNFa, and the S100A8/A9 protein persist at the serum level even after SARS-CoV-2 infection, and this could cause the persistent inflammation present in post-COVID individuals [121]. In another work, serum cytokine levels were assessed in post-COVID individuals, and subjects showing symptoms had higher levels of IL-17 and IL-2, while subjects without alterations had higher levels of IL-10, IL-6, and IL-4 [122]. Moreover, a persistent dysregulation in immune cell subtypes in COVID-19 convalescents up to 24 weeks post-infection was found [123], and the depletion of naive B and T cell subpopulations and the expansion of PD-1 CD8 memory T cells suggest the persistent conversion of naïve T cells to activated states, resulting in the chronic stimulation of the immune response [124]. Finally, the impact of post-COVID conditions using patients’ lived experiences could be used for the development and validation of tools to better understand this complex scenario [71].

### Potential Involvement of Microbiota in Post-Acute COVID-19 Syndrome

Long-lasting symptoms are expected to be an emerging and important concern beyond the early phase of COVID-19 [125]. Particularly important are the neurological symptoms, which can appear during the acute phase of COVID-19 or with a post-infectious delayed onset [22,126,127,128].

Although the treatment of respiratory symptoms has been prioritized in COVID-19 patients, as the respiratory tract is the first target of SARS-CoV-2 infection, and respiratory insufficiency presents the major cause of death, to date, there is increasing evidence that neurological symptoms may also play a significant role in COVID-19.

Importantly, these can be critical in the acute, long COVID, and post-COVID periods [23,129]. The most common neurologic symptoms of COVID-19 include headache, anosmia, and ageusia, but there are numerous reports of encephalopathy, encephalitis, consciousness impairment, and even peripheral nerve disorders [22]. Post-acute COVID-19 is a syndrome that can occur in patients who have had COVID-19 disease, and it is characterized by clinical symptoms that last beyond four weeks after the onset of acute symptoms. The pronounced incidence of neurological symptoms in the post-COVID period has given rise to a new post-COVID-19 neurological syndrome (PCNS), which encompasses the wide variety of symptoms previously mentioned, and further research is needed to define its full clinical picture [23,130].

In a recent study, it was shown that lung function sequelae, such as reduced diffusion lung capacity for carbon monoxide (DLCO), are frequent 3 months after hospital discharge in COVID-19 patients and illustrate that persistent dyspnea is a common clinical marker [131]. Among other neurological symptoms, post-acute COVID-19 individuals also show gastrointestinal alterations (nausea, diarrhea, and abdominal and epigastric pain). It has also been demonstrated that the intestinal microbiomes of patients show higher levels of *Ruminococcus gnavus* and *Bacteroides vulgatus* and lower levels of *Faecalibacterium prausnitzii*. Patients who had no persistent symptoms showed a gut microbiome profile comparable to that of non-COVID-19 controls [132].

## 4. Discussion and Conclusions

A critical role in the clinical outcome of COVID-19 is innate and adaptive immunity dysregulation, suggesting that the severe evolution of COVID-19 is not only driven by exacerbated innate immunity [50,133] but also lymphopenia, along with neutrophil/lymphocyte imbalance [134,135]. A deficient interferon response has also been shown to occur during SARS-CoV-2 infections [136,137]. The tropism of SARS-CoV-2, defined by entry into cells via the widely expressed ACE2 receptor, makes it possible for the infection to spread to different organs, such as the lung, heart, intestine, liver, and brain. This can generate acute as well as chronic damage in various districts [24,25,26,27,28,29,30,31,32,33,34,35,36,37,38,39,40,41,42,43,44,45,46,47,48,49,50,51,52,53,54,55,56,57,58,59,60,61,62,63,64,65,66,67,68,69,70,71,72,73,74,75,76,77,78,79,80,81,82,83,84,85,86,87,88,89,90,91,92,93,94,95,96,97,98,99,100,101,102,103,104,105,106,107,108,109,110,111,112,113,114,115,116,117,118,119,120,121,122,123,124,125,126,127,128,129,130,131,132,133,134,135,136,137,138]. This scenario is also reflected in the composition of the microbiota, the balance of which is regulated by interactions with the immune system. Severe cases of COVID-19 have shown a higher abundance of opportunistic pathogens, and some studies have investigated the contribution of viruses that are present in the microbiota as virome. These studies have investigated virome in COVID-19 patients and determined that the viruses in the guts of these patients are mainly phages (*Myoviridae* virus family (20.91%) and *Siphoviridae* (20.43%)). A potential correlation between virome and bacteriome changes (such as *Tectiviridae* and *Bacteroidaceae*) is also indicated [139]. In a recent study, a significant proportion of eukaryote-associated viruses (with as-yet-undefined taxonomy), along with some plant RNA-related viruses and unclassified viruses, were found in several patients. These analyses suggest that as-yet-unidentified intestinal viruses could influence the host’s immune status and its response to viral infections by determining a more favorable environment for infection by SARS-CoV-2 [140].

The presence of human endogenous retroviruses (HERVs) at the virome level has also recently been detected [141]. HERV sequences are present in our genome and derived from ancestral infections caused by exogenous viruses that infected germ-line cells millions of years ago, at which point they were integrated, fixed, and transmitted to future generations as Mendelian traits [142,143]. Regarding the intestinal tract, the reactivation of HERVs has been documented in colon samples with inflammatory bowel disease, probably induced by Epstein–Barr virus (EBV) infections [144]. Recent works have identified the reactivation of HERVs in COVID-19 patients. In particular, the presence of the HERV-W protein in the blood of COVID-19 patients and its correlation with disease severity and respiratory outcomes has been demonstrated [145]; the presence of HERVs in the bronchoalveolar lavage and respiratory tract has also been assessed [146,147]. A few works have demonstrated the presence of HERVs at the level of the microbiota [148,149], but, considering their potential involvement in inflammatory and pathological processes [150,151], they could represent a potential study and therapeutic target to be considered in virome

Considering the physiological role that the microbiota plays and the very fine balance that is established with the host, SARS-CoV-2 infections cause a real imbalance (Figure 3). Various works have demonstrated an alteration in different tissues, such as oral mucosa, intestinal mucosa, and lung mucosa, and this leads to the presence of an altered microbial population. An emerging and major concern beyond the initial COVID-19 infectious phase is foreseen to result from long-lasting symptoms such as disabling neurological symptoms, among others, which can appear over the course of acute-phase COVID-19 or with a post-infectious delayed onset [126,128,129]. Those various alterations and symptoms are, in some individuals, persistent after infection. In the central nervous system, the pathophysiological mechanisms underlying these symptoms are heterogeneous and may also be related to the excessive cytokine release present in COVID-19 [138], as well as the direct cytopathogenic effects of the virus [152]. Considering the strict association with the gut microbiota (gut–brain axis) [153], this could influence neurological alterations and symptoms. The persistence of these symptoms can result in post-COVID syndrome or long COVID if the infection endures. However, only a few works have found a persistent alteration of the microbiota in post-acute COVID patients, and further studies are needed to elucidate this role.

Based on this evidence, the SARS-CoV-2 virus has a ubiquitous pathogenic potential that causes problems in various organs of the body, occurring early, during, and after the infection due to the onset of chronic inflammation. Regarding the impact of SARS-CoV-2 and variants of concern (VOCs) on global public health, it is necessary to examine the different aspects of host–microbe interaction to limit the broad spectrum of collateral effects due to COVID-19. Considering the upper respiratory tract is the site of the infection and transmission of SARS-CoV-2 and how the microbiota can influence the host response, more studies are needed to associate certain microbial species with the different stages of COVID-19 and to define personalized treatment. Any tissue-specific or microbiota changes must be considered in detail to identify new prognostic and predictive markers of detrimental sequelae and potential targets for therapy.

The possible contribution of dysbiosis occurring in the oropharynx, lungs, and intestines of COVID-19 patients in the manifestation of neurological symptoms, as well as in the persistent inflammation of post-acute infection conditions, should be studied.

## Figures and Tables

**Figure 1 pathogens-11-01198-f001:**
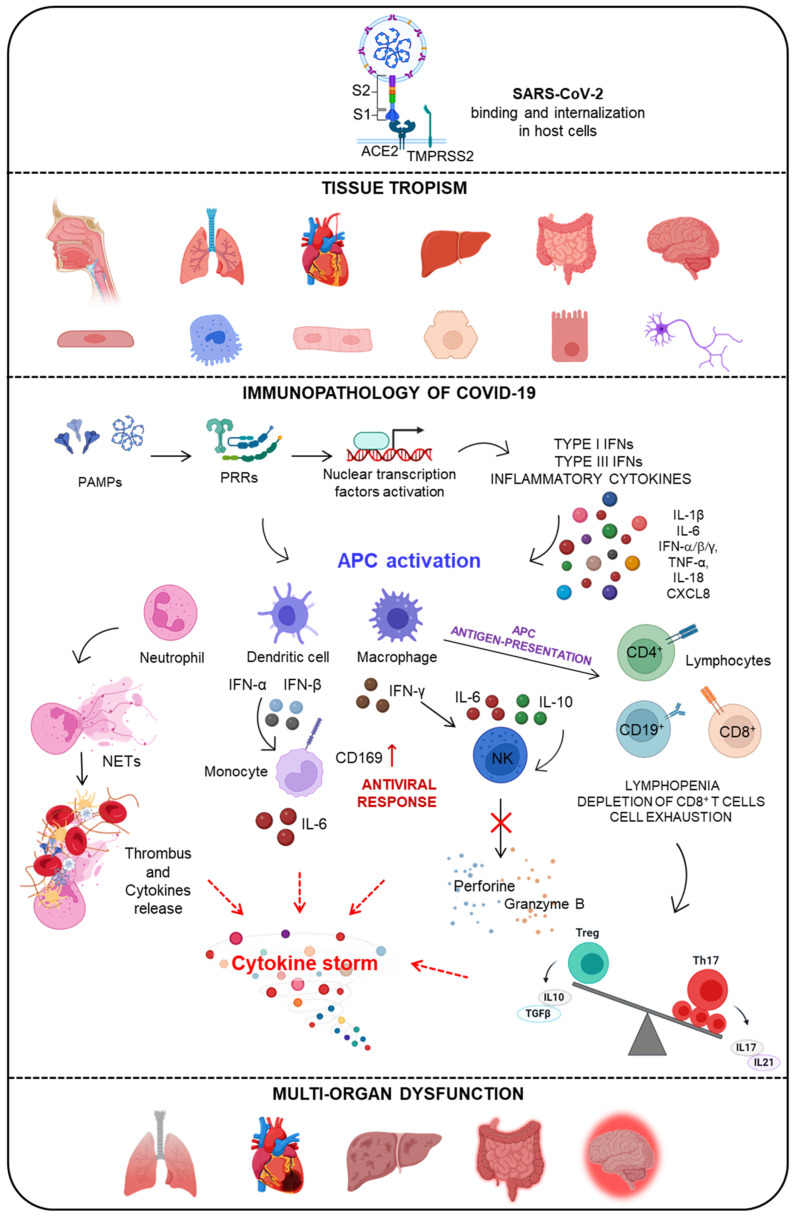
Different tissue tropisms of SARS-CoV-2 and the consequent dysregulated responses in the host. Virus binding and tissue tropism: Entry of SARS-CoV-2 through interaction with the ACE2 receptor, expressed in the host cells in different tissues (respiratory tract, lungs, heart, liver, intestine, and brain) and with enzyme TMPRSS2. Immunopathology of COVID-19 disease: Host cells recognize viral pathogen-associated molecular patterns (PAMPs) through receptors dedicated to recognition and pattern recognition receptors (PRRs) such as Toll-like receptors (TLRs) and RIG-I-like receptors (RLRs). The binding of PAMPs to PRRs triggers the initiation of the inflammatory response against the virus, resulting in the activation of various signaling pathways and subsequent transcription factors. The resulting ‘cytokine storm’ is triggered. Briefly, the infection mechanism results in the activation of mononuclear phagocytes (MPs) such as monocytes, macrophages, and dendritic cells that regulate the first host–pathogen interaction and are antigen-presenting cells (APCs). The release of interferons (IFN-α, IFN-β, INF-γ) from APCs determines the activation of monocytes toward a specific antiviral response with the increase in CD169 expression and the activation of natural killer cells (NK). Moreover, the APCs also determine T cell activation with their maturation. The antiviral response in COVID-19 disease becomes dysregulated due to the excessive release of pro- and anti-inflammatory cytokines. This results in lymphopenia, the depletion of CD4+ and CD8+ cells, and an imbalance of regulatory T cells (Treg) and Th 17. The alteration of neutrophil extracellular traps (NETs), produced to reduce infection, is also established, leading to thrombus formation. Multi-organ dysfunction: The plethora of cellular alterations and immunological hyperactivations are reflected in the different tissues, causing respiratory distress syndrome (ARDS) and systemic multi-organ dysfunction syndrome (MODS).

**Figure 2 pathogens-11-01198-f002:**
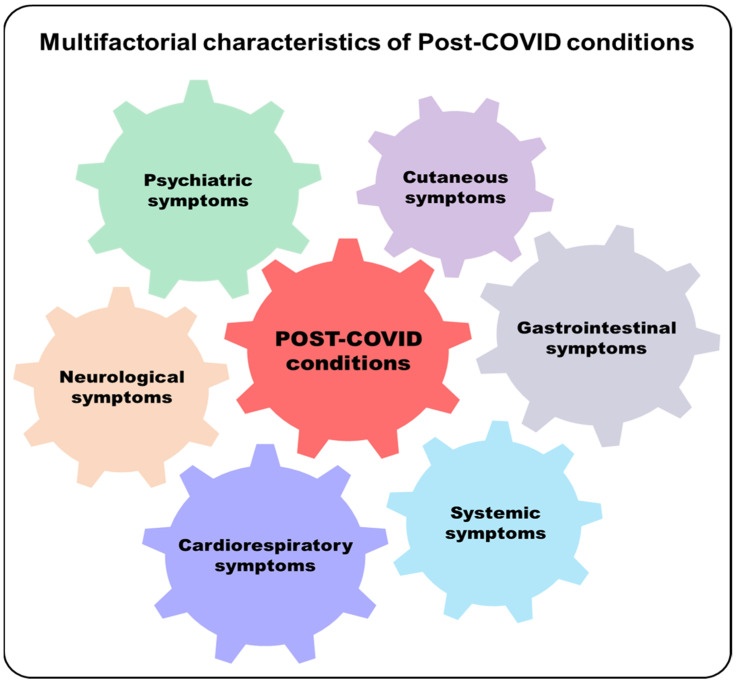
Representation of Post-COVID symptom alterations. Post-COVID conditions include various alterations that result in different types of symptoms such as neurological (cognitive impairment or brain fog, headache, anosmia or dysgeusia, hearing disorder, tremors, paresthesia, vertigo, insomnia and other sleep difficulties, impaired daily function and mobility), psychiatric (mood changes, emotional lability, anxiety, depression), cutaneous (rash (e.g., urticaria), loss of thermoregulation), gastrointestinal (dysbiosis, abdominal pain, diarrhea, nausea, vomiting, loss of appetite), systemic (fatigue, arthralgia, myalgia, fever, pain, menstrual cycle irregularities, erectile dysfunction), and cardiorespiratory (dyspnea, increased respiratory effort, cough, chest pain, palpitations, tachycardia).

**Figure 3 pathogens-11-01198-f003:**
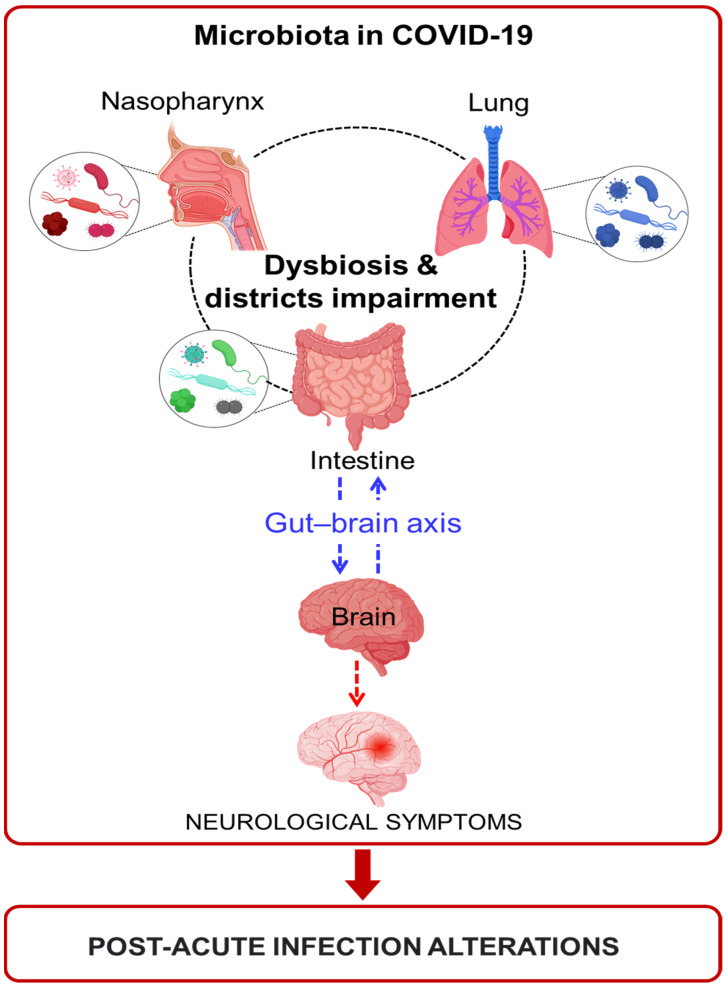
Microbiota alteration during COVID-19 and post-acute infection impairments.

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
