# Peer review of "Multidistrict Host–Pathogen Interaction during COVID-19 and the Development Post-Infection Chronic Inflammation"

_pathogens, 2022, doi:10.3390/pathogens11101198_

Round 1

Reviewer 1 Report

This review manuscript focuses how the invasion of different tissues/organs in humans by SARS-CoV-2 and the changes in microbiota during COVID-19 to reveal the potential pathogenesis of this disease that has caused the pandemic. Although the topic is interesting, there are several major weaknesses significantly diminishing the impact of this manuscript.

1.       Throughout the manuscript, it is difficult to tell whether the authors are interested in the association of dysregulated immunity/hyper-inflammation in severe vs mild COVID, or post-COVID or even long-COVID. It seems that these different stages of disease are considered the same. If it is, please clarify.

2.       Although the authors listed a couple of interesting references to show the microbiota changes, including the increased opportunistic pathogens, in COVID-19 patients, it lacks the clarification that this only occurred in COVID-19 or any systemic infections caused by viruses other than SARS-CoV-2.

3.       It is not surprising to know that SARS-CoV-2 caused reduction of nasopharyngeal commensal bacteria. However, it remains unknow why this is interesting to report or to investigate. Do these changes contribute to spreading of the infection or pathogenesis or just result from a viral infection. If these changes are interesting to pursue, what would be expected to explore, for example, novel therapeutics?

4.       Lines 309-334 described an interesting topic about the mucosal immunity. However, it seems less relevant to the paragraph following this topic in Lines 336-356

Minor comments:

1.       Line 264, does long COVID or post-acute COVID are associated with both pro-inflammatory and anti-inflammatory cytokines? Are there any difference along stages of COVID?

2.       Line 251 what is the MHC-associated pathogen?

3.       Line 252, do CD8 T lymphocytes neutralize infected cells? How?

4.       Does “district” here refer to “tissues” or “organs”?

5.       Line 289 and Line 291 may need further edits.

Author Response

  1. Throughout the manuscript, it is difficult to tell whether the authors are interested in the association of dysregulated immunity/hyper-inflammation in severe vs mild COVID, post-COVID, or even long-COVID. It seems that these different stages of the disease are considered the same. If it is, please clarify.

We thank you for the careful consideration of our manuscript. We noted, as you underlined, that our manuscript lacked a detailed description of the post-COVID condition. For this reason, we have included a dedicated paragraph (paragraph 3 and new figure 2) that should put this point into context.

  1. Although the authors listed a couple of interesting references to show the microbiota changes, including the increased opportunistic pathogens, in COVID-19 patients, it lacks the clarification that this only occurred in -19 or any systemic infections caused by viruses other than SARS-CoV-2.

A new paragraph (paragraph 2.2) entitled “Microbiota alteration during respiratory virus infections” has been added where we contextualize the microbiota in some viral respiratory infections, documented by few papers. We have also included the characteristics of the respiratory tract microbiota under physiological conditions. This is to clarify the involvement, documented by several papers, of microbiota alterations in COVID-19 described in the paragraph (paragraph 2.3) " The contribution of microbiota in COVID-19 disease".

  1. It is not surprising to know that SARS-CoV-2 caused the reduction of nasopharyngeal commensal bacteria. However, it remains unknow why this is interesting to report or to investigate. Do these changes contribute to the spreading of the infection or pathogenesis or just result from a viral infection. If these changes are interesting to pursue, what would be expected to explore, for example, novel therapeutics?

Thank you for your suggestion. Unfortunately, few are the studies that have investigate this point, however we add several annotations throughout the manuscript and have included a comment in the discussion to better highlight this aspect: “Considering that the upper respiratory tract is the site of infection and transmission of SARS-CoV-2 and how much the microbiota can influence the host response, more studies are needed to associate certain microbial species with the different stages of COVID-19 and to define personalized treatment” Lines (602-605 new).

  1. Lines 309-334 described an interesting topic about mucosal immunity. However, it seems less relevant to the paragraph following this topic in Lines 336-356.

As suggested, we have reworded the paragraph 2.3.1 "Microbiota alterations in oral cavity and nasopharynx during COVID-19" as requested. We have also inserted a new paragraph (paragraph 2) to describe the microbiota and its interaction in the immune system (Lines 315-330 new), to better connect the paragraph on the mucosal immune system (paragraph 2.1).

Minor comments:

  1. Line 264, does long COVID or post-acute COVID are associated with both pro-inflammatory and anti-inflammatory cytokines? Is there any difference along the stages of COVID?

We appreciated this question and a description of the cytokines that were detected has been described in the paragraph " Post-acute COVID-19 syndrome". There is still little evidence regarding this type of analysis, but the articles highlighted persistent chronic inflammatory state in post COVID patients.

  1. Line 251 what is the MHC-associated pathogen?

We have included the correct version: “lymphocytes recognize MHC-associated pathogenic peptides via their receptors (TCR). lines (new 264).

  1. Line 252, do CD8 T lymphocytes neutralize infected cells? How?

Specific information on the neutralizing activity of CD8 cells has been added in the text:” In viral infections, CD8+ lymphocytes block infected cells producing effector molecules such as granzyme A and B and perforin or CD95/Fas-mediated apoptosis, while CD4+ follicular helper lymphocytes (Tfh) stimulate B lymphocytes to produce specific antibodies (lines 263-269 new).

  1. Does “district” here refer to “tissues” or “organs”?

We used the term district to define both tissues and organs affected during SARS-CoV-2 infection and their impairment in COVID-19.

  1. Line 289 and Line 291 may need further edits

We have edited the text, as suggested.

Reviewer 2 Report

The review “Multidistrict host-pathogen interaction during COVID-19 and the development post infection chronic inflammation.”

By Fanelli et al is meritorious.

The authors concentrated their attention on 2 points: the role of the immune system and the contribution of microbiota in COVID-19 disease. The latter point has been developed extensively, whereas some more information should be provided for the first one. For example, the role TLR2 has not been addressed at all (Zheng, Min, et al. "TLR2 senses the SARS-CoV-2 envelope protein to produce inflammatory cytokines." Nature immunology 22.7 (2021): 829-838.).

Moreover, I believe that the sentence on Page 2 line  62 A more severe COVID-19 can occur in elderly people with an underlying medical condition such as cardiovascular diseases, lung disease, kidney disease or malignancy, while other severe cases are presumably associated with some genetic predisposition not yet clearly established [9].” is too simplistic and the reference is not appropriate. I agree, and it is well accepted, that as cardiovascular diseases, lung disease, kidney disease or malignancy are the first co-causes of severe COVID. However genetic factors are quite important too. Some polymorphisms that are common in the general population and that affect the functionality, or the expression of the protein encoded by the mutated gene, are associated with severe COVID. In many cases, this observation confirms the role of the affected protein in the pathogenesis of COVID. One very clear example is provided by polymorphisms in TMPRSS2. The authors should state this and cite the literature accordingly for instance:

Dos Santos ACM, Dos Santos BRC, Dos Santos BB, de Moura EL, Ferreira JM, Dos Santos LKC, Oliveira SP, Dias RBF, Pereira E Silva AC, de Farias KF, de Souza Figueiredo EVM. Genetic polymorphisms as multi-biomarkers in severe acute respiratory syndrome (SARS) by coronavirus infection: A systematic review of candidate gene association studies. Infect Genet Evol. 2021 Sep;93:104846. doi: 10.1016/j.meegid.2021.104846. Epub 2021 Apr 30. PMID: 33933633; PMCID: PMC8084602.

Monticelli M, Mele BH, Andreotti G, Cubellis MV, Riccio G. Why does SARS-CoV-2 hit in different ways? Host genetic factors can influence the acquisition or the course of COVID-19. Eur J Med Genet. 2021 Jun;64(6):104227. doi: 10.1016/j.ejmg.2021.104227. Epub 2021 Apr 16. PMID: 33872774; PMCID: PMC8051015.

Di Maria E, Latini A, Borgiani P, Novelli G. Genetic variants of the human host influencing the coronavirus-associated phenotypes (SARS, MERS and COVID-19): rapid systematic review and field synopsis. Hum Genomics. 2020 Sep 11;14(1):30. doi: 10.1186/s40246-020-00280-6. PMID: 32917282; PMCID: PMC7484929.

On page Pag 9 line 345 “In a pre-print work, Budding….”

Budding, Andries, et al. "An age dependent pharyngeal microbiota signature associated with SARS-CoV-2 infection." Available at SSRN 3582780 (2020). should be cited

In general, the paper is easily readable but there are some mistakes. For example:

Line 115 Line 119 errors in punctuation

Several works demonstrate the presence of SARS-CoV-2 in different tissues; a recent study showed the presence of SARS-CoV-2 RNA in the cardiac tissue of 41 cases (43%)                                                                                              out of 95 SARS-CoV-2 positive deceased and how the infection in the heart induced tran-                                                                                  scriptomic alterations associated with the immune response with increased pro-inflam-                                                                                                  matory gene expression. [19].

Page 9 382

“These functions are regulated by the interplay with the immune system and changes in the microbiota composition induced by exogenous viral infection could results in its disruption” RESULT

Page 10 line 422

“Above all are fatigue, fever, respiratory symptoms(difficulty breathing or shortness of breath and cough), joint or muscle pain, loss of smell…..”

 DIFFICULTY BREATHING?

Author Response

  1. The review “Multidistrict host-pathogen interaction during COVID-19 and the development post infection chronic inflammation.”

    By Fanelli et al is meritorious.

    The authors concentrated their attention on 2 points: the role of the immune system and the contribution of microbiota in COVID-19 disease. The latter point has been developed extensively, whereas some more information should be provided for the first one. 
    the role TLR2 has not been addressed at all (Zheng, Min, et al. "TLR2 senses the SARS-CoV-2 envelope protein to produce inflammatory cytokines." Nature immunology 22.7 (2021): 829-838.

Thank you for your suggestion. we have included the required information for TLR2 with this reference and another about the role of TLR2 in lung inflammation. (New Lines 178-180).

  1. Moreover, I believe that the sentence on Page 2 line  62 “ A more severe COVID-19 can occur in elderly people with an underlying medical condition such as cardiovascular diseases, lung disease, kidney disease or malignancy, while other severe cases are presumably associated with some genetic predisposition not yet clearly established [9].” is too simplistic and the reference is not appropriate. However genetic factors are quite important too. Some polymorphisms that are common in the general population and that affect the functionality, or the expression of the protein encoded by the mutated gene, are associated with severe COVID. In many cases, this observation confirms the role of the affected protein in the pathogenesis of COVID. One very clear example is provided by polymorphisms in TMPRSS2.”The authors should state this and cite the literature accordingly for instance: Dos Santos ACM, Dos Santos BRC, Dos Santos BB, de Moura EL, Ferreira JM, Dos Santos LKC, Oliveira SP, Dias RBF, Pereira E Silva AC, de Farias KF, de Souza Figueiredo EVM. Genetic polymorphisms as multi-biomarkers in severe acute respiratory syndrome (SARS) by coronavirus infection: A systematic review of candidate gene association studies. Infect Genet Evol. 2021 Sep;93:104846. doi: 10.1016/j.meegid.2021.104846. Epub 2021 Apr 30. PMID: 33933633; PMCID: PMC8084602. Monticelli M, Mele BH, Andreotti G, Cubellis MV, Riccio G. Why does SARS-CoV-2 hit in different ways? Host genetic factors can influence the acquisition or the course of COVID-19. Eur J Med Genet. 2021 Jun;64(6):104227. doi: 10.1016/j.ejmg.2021.104227. Epub 2021 Apr 16. PMID: 33872774; PMCID: PMC8051015. Di Maria E, Latini A, Borgiani P, Novelli G. Genetic variants of the human host influencing the coronavirus-associated phenotypes (SARS, MERS and COVID-19): rapid systematic review and field synopsis. Hum Genomics. 2020 Sep 11;14(1):30. doi: 10.1186/s40246-020-00280-6. PMID: 32917282; PMCID: PMC7484929.

We agree with these considerations, it is well accepted, that cardiovascular diseases, lung disease, kidney disease or malignancy are the first co-causes of severe COVID-19 as well as a genetic predisposition. We have changed this concept in the text and added some information on TMPRSS2 and IFN polymorphisms with also the references suggested. (new lines 45-75).

  1. On page Pag 9 line 345 “In a pre-print work, Budding….”Budding, Andries, et al. "An age-dependent pharyngeal microbiota signature associated with SARS-CoV-2 infection." Available at SSRN 3582780 (2020). should be cited

We have modified and added the reference

  1. In general, the paper is easily readable but there are some mistakes. For example:
  1. A) Line 115 Line 119 errors in punctuation

We have checked and modified

  1. B) Page 9 382 “These functions are regulated by the interplay with the immune system and changes in the microbiota composition induced by exogenous viral infection could results in its disruption”

We have modified

  1. Page 10 line 422 “Above all are fatigue, fever, respiratory symptoms (difficulty breathing or shortness of breath and cough), joint or muscle pain, loss of smell…..” DIFFICULTY BREATHING?

Thank for your careful revision. we have corrected the mistakes in the text.

Reviewer 3 Report

Comments for the authors

SARS-CoV-2 infection usually leads to cytokine storm, lymphopenia, and cellular exhaustion and thus ARDS and MODS are established in severe patients. Composition of the microbiota is also largely influenced, the balance of which is regulated by the interaction with the immune system. A change in microbial diversity has been demonstrated in COVID-19 patients compared to healthy donors, with an increase in potentially pathogenic microbial genera. In addition to other symptoms, particularly the neurological, the occurred dysbiosis persists after SARS-CoV-2 infection determining the post-acute COVID syndrome. Based on the above features of SARS-CoV-2, the authors described and contextualised the role of the immune system unbalance and dysbiosis during SARS-CoV-2 infection, from the acute to the post-COVID-19. Gut-brain axis, as one of the analysis of new multidistrict parameters could also be applied to address the chronic multisystem dysfunction related to COVID-19.

Major concerns:

Figure2, There is no figure legend, the figure 2 title is" Microbiota alteration during COVID-19 and Post-acute infection impairments", however, the authors just  emphasize the neurologic symptoms, missing the impairment of nasopharynx, lung and intestine, in order to be consistent”nasopharynx symptoms, lung symptoms and intestine symptoms” would be better. And it would also be better to compare the composition of microbiota in nasopharynx, lung and intestine between the healthy people and post-acute infection patients shown in table.

Minor concerns:

1.        line 499, " alteration of the microbiota in post-acute COVID patients and further studies are need to", need” should be “ needed”.

2.        line 473, “such” should be “such as”

3.        line 283, “subsequently” should be “subsequent”,

4.        line 371-372, “endotoxin” should be “endotoxins”, or “affect” should be “affects”.

5.        line 481, There should be a comma before " but".

6.        In the Figure1, “ bingind” should be “binding”,

7.        In the Figure1,  the Type I IFNs IFN-α and IFNβ-are not listed there, while Type III INFS and INFLAMMATORY CYTOKINES are listed.

Author Response

Reviewer 3.

  1. Figure2, There is no figure legend, the figure 2 title is" Microbiota alteration during COVID-19 and Post-acute infection impairments" . however, the authors just emphasize the neurologic symptoms, missing the impairment of nasopharynx, lung and intestine, to be consistent” nasopharynx symptoms, lung symptoms and intestine symptoms” would be better.

Thank for reporting the absence of the legend. We have included the legend in the text: “Possible contribution of dysbiosis occurring in the oropharynx, lungs, and intestines of COVID-19 patients in the manifestation of neurological symptoms and persistent inflammation of post-acute infection conditions.”

We also modified the figure by adding the "district impairments" after dysbiosis to highlight this aspect of the global changes that occur in the acute phase and could contribute to the development of the post-acute conditions.

  1. And it would also be better to compare the composition of microbiota in nasopharynx, lung and intestine between the healthy people and post-acute infection patients shown in table.

It would be interesting to explore this further because there is currently little evidence of this comparison. The purpose of this review is the promotion of these studies by considering the impact of post-COVID conditions in the population and specially to include the microbiota in the overall assessment of a disease, particularly in COVID-19 and its long-term repercussions. In the final part of the discussion, these concepts were added. (new lines 599-609).

Minor concerns:

  1. line 499, " alteration of the microbiota in post-acute COVID patients and further studies are need to", “need” should be “needed”.
  2. line 473, “such” should be “such as”
  3. line 283, “subsequently” should be “subsequent”,
  4. line 371-372, “endotoxin” should be “endotoxins”, or “affect” should be “affects”.
  5. line 481, There should be a comma before " but".
  6. In the Figure1, “ bingind” should be “binding”,
  7. In Figure1, the Type I IFNs IFN-α and IFNβ-are not listed here, while Type III INFS and INFLAMMATORY CYTOKINES are listed.

Thank you for your careful review. We have edited the text and corrected the errors.

Round 2

Reviewer 1 Report

No more comments.

Reviewer 2 Report

The paper was revised according to reviewers' suggestions and can be published in the present form